# Model-based translation of DNA damage signaling dynamics across cell types

**Muriel M. Heldring**, **Lukas S. Wijaya**, **Marije Niemeijer**, **Huan Yang**, **Talel Lakhal**, **Sylvia E. Le Dévédec**, **Bob van de Water**, **Joost B. Beltman***

Division of Drug Discovery and Safety, Leiden Academic Centre for Drug Research, Leiden University, Leiden, The Netherlands

* j.b.beltman@lacdr.leidenuniv.nl

**Data Availability Statement:** Data, models and code is available via the persistent link https://doi.org/10.5281/zenodo.6458438.

## Abstract

Interindividual variability in DNA damage response (DDR) dynamics may evoke differences in susceptibility to cancer. However, pathway dynamics are often studied in cell lines as alternative to primary cells, disregarding variability. To compare DDR dynamics in the cell line HepG2 with primary human hepatocytes (PHHs), we developed a HepG2-based computational model that describes the dynamics of DDR regulator p53 and targets MDM2, p21 and BTG2. We used this model to generate simulations of virtual PHHs and compared the results to those for PHH donor samples. Correlations between baseline p53 and p21 or BTG2 mRNA expression in the absence and presence of DNA damage for HepG2-derived virtual samples matched the moderately positive correlations observed for 50 PHH donor samples, but not the negative correlations between p53 and its inhibitor MDM2. Model parameter manipulation that affected p53 or MDM2 dynamics was not sufficient to accurately explain the negative correlation between these genes. Thus, extrapolation from HepG2 to PHH can be done for some DDR elements, yet our analysis also reveals a knowledge gap within p53 pathway regulation, which makes such extrapolation inaccurate for the regulator MDM2. This illustrates the relevance of studying pathway dynamics in addition to gene expression comparisons to allow reliable translation of cellular responses from cell lines to primary cells. Overall, with our approach we show that dynamical modeling can be used to improve our understanding of the sources of interindividual variability of pathway dynamics.

## Author summary

Susceptibility to develop cancer varies among people, partially due to differences in genetic background. Ideally, healthy human-derived cells are used to investigate intracellular signaling pathways and their interindividual variability contributing to cancer susceptibility. Because cells from healthy human tissue are difficult to obtain and culture for periods longer than a few days, cell lines are often used as substitute. However, it is unclear to what extent signaling dynamics in cell lines represent dynamics in healthy human tissue. We asked whether we could reproduce interindividual variability in DNA damage

**Funding:** This work has received funding from the European Union's Horizon 2020 research and innovation programme under grant agreement No. 681002 (BVDW and JBB) (EU-ToxRisk) and the ZonMW InnoSysTox program under grant agreement No. 40-42600-98-14030.(BVDW and JBB) Moreover, this work has received funding from the TransQST and eTRANSAFE projects, which have both received support from IMI2 Joint Undertaking under Grant Agreements No. 116030 (BVDW) and No. 777365(BVDW), respectively. This Joint Undertaking receives support from the European Union's Horizon 2020 research and innovation programme and the European Federation of Pharmaceutical Industries and Associations (EFPIA). This manuscript reflects only the authors' view and IMI JU is not responsible for any use that may be made of the information it contains. The funders had no role in study design, data collection and analysis, decision to publish, or preparation of the manuscript.

**Competing interests:** The authors have declared that no competing interests exist.

response gene expression in a set of 50 human liver cell donors. Therefore, we built a mathematical model that simulates temporal expression dynamics of the DNA damage response in the HepG2 liver cell line upon chemical activation and used the simulations to create virtual donors. Our virtual donors displayed similar relations between genes as the samples from human donors, provided that we adjusted the strength of specific molecular interactions. Thus, our approach can be used to examine the applicability of widely used cell systems to healthy human tissue in terms of their dynamic responses.

## Introduction

An effective cellular response to DNA damage after exposure to genotoxic chemicals is critical to prevent cancer development [1–3]. However, the response to chemicals and susceptibility to chemical-induced carcinogenesis is highly variable across the human population [4]. This variability is in part caused by well-studied factors such as interindividual genomic variation and differences in gene and protein expression levels that influence processes such as drug metabolism [5,6], detoxification [7] and DNA repair [8,9]. To understand how gene and protein expression patterns determine these processes, insight into the underlying DNA damage pathway dynamics is needed. However, time-resolved protein expression data in human primary cells cannot be easily obtained due to technical limitations, such as their limited availability [10–13]. Therefore, cell lines are often used as experimental model systems representative for healthy human tissue. However, it is unclear whether experiments performed using cell lines are sufficient to quantitatively predict dynamics in primary human cells.

Examining DDR pathway dynamics in liver tissue is of special interest, because the liver is the primary site for drug metabolism [14,15] and is therefore often a target for carcinogenesis [16,17]. A frequently used experimental model system for primary human hepatocytes (PHHs) is the hepatocellular carcinoma cell line HepG2. Wink et al. (2017) created a HepG2-GFP (green fluorescent protein) reporter assay that monitors the expression dynamics of transcription factor p53 and its downstream targets MDM2, p21 and BTG2, four essential proteins in the DDR pathway [18–20]. The central regulator of the DDR, tumor suppressor protein *TP53*/p53, is phosphorylated by DNA damage sensing kinases ataxia-telangiectasia mutated (ATM), ATM and Rad3-related (ATR) and DNA-dependent protein kinase catalytic subunit (DNA-PKcs), and their downstream Checkpoint kinases 1 and 2 (CHEK1 and CHEK2) in reaction to DNA insults [21–25]. Phosphorylated p53 (p53-p) activates DNA repair mechanisms such as nucleotide excision repair (NER) that removes adducts and restores DNA [26]. In addition, activated p53 translocates to the nucleus and is likely responsible for the transcriptional activation of hundreds of downstream targets [27,28]. Among the verified p53-regulated proteins is *MDM2*/MDM2, that negatively regulates p53 by ubiquitination of its carboxy terminus [29]. In addition, p53 induces several proteins that play key roles in DNA damage repair, cell cycle arrest, senescence and apoptosis. Among these proteins are *CDKN1A*/p21 and *BTG2*/BTG2, that are both involved in the regulation of cell cycle arrest [30–33]. In case of severe DNA damage, cell cycle arrest through p53-dependent activation is essential to prevent proliferation of damaged cells and to provide time for repair and recovery. Thus, p53 and its downstream targets fulfill an essential role in the recovery of the instability created by DNA damage.

Here, we used previously established protein expression data obtained by employing HepG2 p53-GFP, MDM2-GFP, p21-GFP and BTG2-GFP reporter cells [34] and gene expression data in HepG2 cells and PHHs from 50 donors [35] to study the similarity between these

cell types in their response to the DNA damaging compound cisplatin. Specifically, we examined the predictive capacity of *TP53* expression levels on *MDM2*, *CDKN1A* and *BTG2* downstream target expression to investigate whether interindividual variability in basal p53 gene expression is predictive for the response to cisplatin. To this purpose, we calibrated a newly developed dynamic computational model to the protein expression data. Considering this model to be representative for PHHs, we generated virtual donor samples and compared the model-predicted correlations between basal *TP53* expression and its downstream targets *MDM2*, *CDKN1A* and *BTG2* in virtual samples with the correlations observed in 50 PHH donor samples. The HepG2-based virtual donors could well explain the moderately positive correlations observed among PHHs, but not the negative *TP53-MDM2* correlation. We show that parameter alterations that affect p53 and MDM2 dynamics are not sufficient to explain the observed *TP53-MDM2* relationship. Because our model does accurately describe HepG2 protein expression, this implies there is a knowledge gap in the *TP53-MDM2* relation for quantitative extrapolation of HepG2-based data to responses of primary liver cells. With the presented approach, we provide a new way to investigate the applicability of cell line responses to those of primary cells, while taking the interindividual variability among primary cell donors into account.

## Methods

### Experimental details

**Cell culture.**   Human hepatoma (HepG2) cells were purchased from ATCC (clone HB8065) and maintained in DMEM high glucose (Fisher Scientific) supplemented with 10% (v/v) FBS (Fisher Scientific), 250 U/ml penicillin and 25 μg/ml streptomycin (Fisher Scientific) in humidified atmosphere at 37 degrees Celsius and 5% $CO_2$/air mixture. All the BAC-GFP HepG2 reporter cell lines (p53-GFP, MDM2-GFP, and p21-GFP) were previously established and characterized [18]. The cells were used between passage 14 and 20 and seeded in Greiner black μ-clear 384 well plates, at 8000 cells per well.

**Immunostaining.**   HepG2-WT cells were plated in 96 well plates with a micro clear bottom with a density of 32,000 cells/well and exposed to 2.5, 5, 10 and 25 μM cisplatin (Ebewe). Following an exposure of 1, 2, 3, 6, 8, 16 or 24 hours, we fixed the cells with 1% formaldehyde/ 0.1% Triton X100 (Sigma-Aldrich) incubation for 15 minutes. Next, we incubated the cells for 48 hours at 4˚C with a Phospho-Histone H2A.X (Ser139) Antibody (Cell Signalling Technology, 9718T) with a 1:800 dilution in 0.5% BSA (Sigma-Aldrich) in PBS (Sigma-Aldrich) and subsequently for 1 hour with an Alexa Fluor 647-conjugated AffiniPure Goat Anti-Rabbit IgG (Jackson ImmunoResearch) with a dilution factor of 1: 250 in 0.5% BSA in PBS. The nuclei were stained with a Hoechst solution with a dilution factor of 1:1000 in PBS for 15 minutes. The imaging was done with 20x magnification objective.

**Western blot.**   HepG2-WT cells were plated in 6 well plates with a density of 1,250,000 cells/well and exposed to 1, 2.5, 5, 10, 15, 20, 25 and 50 μM cisplatin (Ebewe). After exposure, we lysed the cells with RIPA buffer, containing a Protease Inhibitor Cocktail and sodium fluoride (Sigma-Aldrich), which both had a dilution factor of 100. Subsequently, the lysate was sonicated and mixed with a loading buffer, containing 2-Mercaptoethanol (Acros Organics), in a ratio of 6:1. A 10% acrylamide running gel was used for p53, p53-S15 and p53-S46 separation and a 15% acrylamide running gel for p21 separation. The gel electrophoresis was performed under a voltage set at 55 V for 30 minutes, after which the voltage was increased to 110 V for an additional 60 minutes. We performed blotting at a voltage of 100 V for 2 hours with PVDF membranes which we blocked in 5% BSA in Triss-Buffered Saline with 0.005% Tween 20 (TBS-T) for 1 hour at room temperature. We incubated the membranes overnight with

primary antibodies (Cell Signalling Technology) for total p53 (9282), p53-S15 (9284), p53-S46 (2521) and p21 (2947) with a dilution factor of 1000 in TBS-T at 4°C, and the next day we incubated them with a Peroxidase-conjugated AffiniPure Goat Anti-Rabbit IgG secondary antibody (Jackson ImmunoResearch) with a dilution factor of 3000 in TBS-T for 1 hour at room temperature. The tubulin bands were identified with a Monoclonal Anti-α-Tubulin antibody (Sigma-Aldrich) and an Alexa Fluor 647-conjugated AffiniPure Goat Anti-Mouse IgG (Jackson ImmunoResearch) for respectively 1 hour and 30 minutes at room temperature. We assessed protein formation with the Amersham Imager 600 (GE Healthcare Bio-Sciences AB).

**TempO-Seq analysis.**    To evaluate variability in p53 signaling, available TempO-seq data of a panel of 54 different PHHs and wild type HepG2 (HepG2-WT) cells exposed to a wide concentration range of cisplatin for 8 and 24 h was used [35]. In short, plateable cryopreserved PHHs from 54 individuals (KaLy-Cell, Plobsheim, France) and HepG2-WT cells were plated at a density of 70.000 cells per well in 96 wells BioCoat Collagen I Cellware plates (Corning, Wiesbaden, Germany). This procedure was repeated three times to capture technical variability. After 24h attachment, PHHs were exposed with cisplatin (Ebewe) for 8 and 24 h and lysed with 1x TempO-seq lysis buffer (BioSpyder). Lysates were stored at -80°C and shipped for TempO-seq analysis using the S1500+ gene set [36] at BioSpyder (Carlsbad, CA, USA).

**Preprocessing and analysis of TempO-Seq data.**    We excluded four PHH donor samples from analysis due to non-confluency, leaving 50 donor samples for analysis. The gene counts for each individual sample were added to obtain library sizes. Samples with a library size smaller than 100,000 were excluded from further analysis (S1A Fig). Genes with zero counts in more than 10% of all samples were removed from the data set. Raw gene counts were normalized through scaling by size factors [37] with the DESeq2 counts function by specifying normalized = TRUE from the DESeq2 package [38] in R version 4.0 [39]. The normalized counts were subsequently $log_2$ transformed using an offset of 1, i.e., $log_2$ (count + 1). We checked for outlying donor samples and differences in gene expression of the entire S1500 + gene set by performing dimensionality reduction with PCA analysis and subsequently t-distributed Stochastic Neighbor Embedding (t-SNE) analysis [40] on the first 32 principal components and with the perplexity set at 22. Gene expression heatmaps for all concentrations and time points were generated based on the mean of the $log_2$ normalized data for three biological replicates. We used the mean of the three available *MDM2* probes to get a single count per sample for *MDM2*. Thereafter, hierarchical clustering was performed on the Euclidean distance matrix using complete linkage. We fitted dose-response curves through the expression data up to and including the 10 μM dosage per cell type using a Hill equation with a 4-parameter log-logistic model using the drc library in R.

**Differential gene expression and functional enrichment analysis.**    Differentially expressed genes analysis was performed with DESeq2. A design classifier specifying the combination of compound, concentration and timepoint was used as design formula in the DESeq-DataSet object. DESeq2 results, performed separately for PHH and HepG2 samples, were extracted per design classifier using the result function with alpha = 0.05, and uploaded to the PHH TXG-MAPr [41]. This method arranges genes in predefined modules of co-regulated genes and calculates eigengene scores (EGSs) for modules that represent their (de)activation. We analysed the resulting modules with the 20 highest positive EGSs for overlap between HepG2 cells and PHHs with a Venn diagram. Besides the TXG-MAPr analysis, genes with an adjusted *p*-value < 0.01 and $log_2$ fold change > 0.1 were used for functional enrichment analysis. Since the adjusted *p*-values are set to NA when rows have low mean normalized counts, these genes were excluded. Functional enrichment was done using the gost function from g:Profiler [42] with a custom background set of all genes in the S1500+ gene set [36]. The false discovery rate method was used to correct for multiple testing. We manually

selected cytotoxicity and cell health Gene Ontology (GO) terms of interest to study the functional enrichment for these terms. Moreover, we selected the 10 most significantly enriched terms per concentration at the 8-hour time point.

**Live-cell imaging of HepG2 BAC-GFP reporters.** We used protein expression data from a previously established data set [34]. To obtain the data utilized in our analysis, the nuclei of HepG2 BAC-GFP reporter cells were stained with 100 ng/ml live Hoechst 33342 (1:10000 dilution) in complete DMEM high glucose for 2 hours. Thereafter, the medium/Hoechst mixture was refreshed with complete DMEM containing 0.2 μM propidium iodide (PI) and Annexin-V-Alexa633 (AnV), and cisplatin (Ebewe) at 1, 2.5, 5, 10, 15, 20, 25 and 50 μM was added. This procedure was repeated three to four times to capture biological variability. We imaged the plates starting within one hour after cisplatin exposure up to 65–72 hours using a Nikon TiE2000 confocal laser microscope (laser excitation wavelengths: 647 nm, 540 nm, 488 nm, and 408 nm), equipped with automated stage and perfect focus system. During imaging, the plates were maintained in humidified atmosphere at 37 degrees Celsius and 5% $CO_2$/air mixture. Imaging was done with 20x magnification objective and performed every 1.5 hours.

**MDM2 feedback disruption with Nutlin.** We used the same procedure as described above to generate new live-cell imaging data of HepG2 BAC-GFP reporters, aiming to examine the effect of MDM2 feedback disruption. For the condition with MDM2 inhibition, we added Nutlin (Sigma, N6287; final concentration 10 μM) to complete DMEM and started imaging within one hour after exposure. We imaged in the same manner as previously explained, and we monitored p53- and MDM2-GFP abundance with a 1.5-hour interval for 72 hours.

**Cisplatin re-exposure.** Following 72 hours of imaging in a HepG2 BAC-GFP reporter experiment as described above, we extracted the medium containing remaining cisplatin from each well and we used this medium to expose fresh HepG2 p53-GFP cells that were seeded 3 days prior to this exposure in new 384 well plates. The seeding procedure and timing were identical as in the original exposure, thus resulting in similar cell counts at the day of exposure. Moreover, we imaged in the same manner as previously explained, and we monitored p53-GFP dynamics with a 2-hour interval for 24 hours as an indicator for the activity of remaining cisplatin after several days of previous exposure.

**Image and data analysis.** We used quantified single cell data sets from Wijaya et al. (2021) as starting point. To quantify the GFP intensity of the new live-cell imaging data, we used the in-house WMC Segment plug-in [43] in ImageJ to perform segmentation of cell nuclei in the images. Foci segmentation for γ-H2AX was done using ImageJ with the Subtract Background process (rolling ball radius = 6), a Gaussian Blur (sigma = 1) and the FociPicker3D plugin [44]. We set the minISetting to 0.55, the ToleranceSetting to 1000, the minimum pixels for foci to 4 and the FociShapeR to 6, whereas all other settings were kept at their default values. The resulting binary images were loaded into CellProfiler version 3.1.9 together with the raw images. To quantify GFP intensity (representative for protein expression) within segmented nuclei and associated cytoplasmic compartments, we used the IdentifyPrimaryObjects, IdentifySecondaryObjects, IdentifyTertiaryObjects and OverlayOutlines modules. Modules MeasureObjectSizeShape and MeasureObjectIntensity subsequently measured the integrated intensity in those regions. Since the GFP expression in single cells was not normally distributed, the geometric mean over all cells per image was calculated to obtain population level measurements. Images that exhibited a clear deviation from the general trend because of extremely low cell density based on visual inspection were excluded from the analysis. Thereafter, we calculated the average of the two technical replicates, i.e., two images from the same well. To prepare the data from Wijaya et al. (2021) for model fitting, we performed background correction per plate by subtracting the average of the GFP signal in the DMEM control condition per timepoint from the average GFP value per timepoint in the cisplatin conditions.

In addition, we applied min-max normalization per plate to rescale the intensities between 0 and 1, with function $\vec{x}' = \frac{x_i - \min(\vec{x})}{\max(\vec{x}) - \min(\vec{x})}$, where $x_i$ with $i \in \{1,2,3,\ldots,n\}$ represents each single GFP measurement and $\vec{x}$ represents all $n$ GFP measurements within one imaging experiment. The timepoints of measurements were not identical across individual biological replicates, since the time between exposure and imaging differed slightly between experiments. To align the measurement time points among biological replicates, we interpolated the data with the B-spline function `bs` in R using 6 degrees of freedom and a third-degree polynomial. Subsequently, we determined the interpolated GFP expression starting at the 1-hour and ending at the 65-hour timepoints with an interval of 1.5 hours. We determined maximum values of the interpolated data for all four proteins and defined the peak delay as the difference in the timepoint of maximum GFP expression of downstream targets and the mean of the timepoints at which maximal p53-GFP values were reached. The response latency per time-response curve was determined by taking the first timepoint at which the mean protein expression across replicates was higher than 1.5 times the mean expression in control condition. For the proportion of PI- and AnV-positive cells, we used the PI/nuclei and AnV/nuclei overlap readout and considered a cell positive if this fraction was larger than 0.1.

## Computational modeling

**Modeling.**   We built a mathematical model to describe the experimentally observed protein dynamics. We used a set of ten ordinary differential equations (ODEs) to simulate the dynamics of the state variables DNA damage, p53 mRNA, p53, phosphorylated p53, MDM2 mRNA and protein, p21 mRNA and protein, and BTG2 mRNA and protein, respectively denoted by $DD$, $P53_{RNA}$, $P53$, $P53_p$, $MDM2_{RNA}$, $MDM2$, $P21_{RNA}$, $P21$, $BTG2_{RNA}$ and $BTG2$. All ODEs were based on commonly used mathematical terms for biochemical reactions, such as mass-action and Hill kinetics [45,46]. We modeled the cisplatin concentration $S$ that leads to DNA damage as an explicit function of time $t$ by:

$$S(t) = EC_i \cdot e^{-\tau \cdot t}. \qquad \text{Eq 1}$$

Here, $EC_i$ is the effective cisplatin concentration causing DNA damage at applied concentration 1 ($i = 1$), 2.5 ($i = 2$) and 5 μM ($i = 3$), and $\tau$ is the cisplatin decay rate caused by the combined effect of cellular metabolism, chemical interactions with intra- and extracellular components and plastic binding. The $EC_1$ parameter describing the effective concentration originating from 1 μM applied cisplatin concentration was fixed to 1. The rate of change in DNA damage over time depends on the constitutive DNA damage occurrence rate ($ks_{DD}$), the cisplatin concentration ($S$), and the repair rate ($kd_{DD}$) stimulated by $P53_p$:

$$\frac{dDD}{dt} = ks_{DD} - kd_{DD} \cdot DD \cdot P53_p + S. \qquad \text{Eq 2}$$

The dynamics of p53 mRNA, the two p53 protein species and the downstream targets are denoted in Eqs 3–11. We describe $P53_{RNA}$ and $P53$ by basal synthesis and degradation rates ($ks_{p53\ RNA}$, $ks_{p53}$, $kd_{p53\ RNA}$, and $kd_{p53}$ respectively). The equations for the p53 protein species describe additional dephosphorylation and DNA damage-dependent phosphorylation processes with rate parameters $k_{dp}$ and $k_p$, and MDM2-dependent degradation at rates $kd_{p53\ mdm2}$ and $kd_{p53p\ mdm2}$ for $P53$ and $P53P$ respectively. The ODEs describing the mRNA and protein dynamics of MDM2, p21, BTG2 all have the same form, with basal synthesis ($ks_{mdm2\ RNA}$, $ks_{mdm2}$, $ks_{p21\ RNA}$, $ks_{p21}$, $ks_{btg2\ RNA}$, $ks_{btg2}$) and degradation rates ($kd_{mdm2\ RNA}$, $kd_{mdm2}$, $kd_{p21\ RNA}$, $kd_{p21}$, $kd_{btg2\ RNA}$, $kd_{btg2}$), and a phosphorylated p53-dependent mRNA induction. The

latter is modelled as Hill equation with power 4, since phosphorylated p53 binds to the DNA as tetramer. Note that we also attempted replacing the power of 4 in the Hill equations of the ODEs for MDM2, p21 and BTG2 mRNA to power 1 to investigate how this model change affected the expected dynamics. Parameters $ks_{mdm2\ p53p}$, $ks_{p21\ p53p}$, $ks_{btg2\ p53p}$ describe the maximal p53-p dependent protein synthesis rates and $Km_{mdm2}$, $Km_{p21}$, $Km_{btg2}$ are the p53-p concentrations at which the reaction rates are half-maximal. The remaining ODEs thus become:

$$\frac{dP53_{RNA}}{dt} = ks_{p53\ RNA} - kd_{p53\ RNA} \cdot P53_{RNA}, \quad\quad \text{Eq 3}$$

$$\frac{dP53}{dt} = ks_{p53} \cdot P53_{RNA} + k_{dp} \cdot P53_p - k_p \cdot P53 \cdot DD - kd_{p53} \cdot P53 - kd_{p53\ mdm2} \cdot P53 \\ \cdot MDM2, \quad\quad \text{Eq 4}$$

$$\frac{dP53p}{dt} = k_p \cdot P53 \cdot DD - k_{dp} \cdot P53_p - kd_{p53p} \cdot P53_p - kd_{p53p\ mdm2} \cdot P53_p \cdot MDM2, \quad\quad \text{Eq 5}$$

$$\frac{dMDM2_{RNA}}{dt} = ks_{mdm2\ RNA} + \frac{ks_{mdm2\ p53p} \cdot P53_p{}^4}{Km_{mdm2}^4 + P53_p{}^4} - kd_{mdm2\ RNA} \cdot MDM2_{RNA}, \quad\quad \text{Eq 6}$$

$$\frac{dMDM2}{dt} = ks_{mdm2} \cdot MDM2_{RNA} - kd_{mdm2} \cdot MDM2, \quad\quad \text{Eq 7}$$

$$\frac{dP21_{RNA}}{dt} = ks_{p21\ RNA} + \frac{ks_{p21\ p53p} \cdot P53_p{}^4}{Km_{p21}^4 + P53_p{}^4} - kd_{p21\ RNA} \cdot P21_{RNA}, \quad\quad \text{Eq 8}$$

$$\frac{dP21}{dt} = ks_{p21} \cdot P21_{RNA} - kd_{p21} \cdot P21, \quad\quad \text{Eq 9}$$

$$\frac{dBTG2_{RNA}}{dt} = ks_{btg2\ RNA} + \frac{ks_{btg2\ p53p} \cdot P53_p{}^4}{Km_{btg2}^4 + P53_p{}^4} - kd_{btg2\ RNA} \cdot BTG2_{RNA}, \text{ and} \quad\quad \text{Eq 10}$$

$$\frac{dBTG2}{dt} = ks_{btg2} \cdot BTG2_{RNA} - kd_{btg2} \cdot BTG2. \quad\quad \text{Eq 11}$$

In addition to Eqs 1–11, we used observable functions to map the state variables to the experimental observables, according to the method described in [47]. Since the GFP-tag should be equally present on both p53 species, the sum of p53 and its phosphorylated form represents the observed p53-GFP intensity in the cells and is therefore described by:

$$P53^O = scaling_{p53} \cdot \left( P53 + P53_p \right) + offset_{p53}. \quad\quad \text{Eq.12}$$

MDM2, p21 and BTG2 model states were mapped to the observables with:

$$Y^O = scaling_Y \cdot Y + offset_Y, \qu\quad\quad \text{Eq.13}$$

where $Y \in \{MDM2, P21, BTG2\}$. Besides the above described DDR model, we also investigated a model with a non-linear p53-MDM2 feedback, and an alternative DDR model that included binding of phosphorylated MDM2 to p53 mRNA (see S1 Methods).

**Model parameterization.** To perform minimization of the cost function we used the trust region approach within the nonlinear least-squares solver of the scipy package in Python version 3.7.2. The efficiency of minimization was improved by providing sensitivity equations and steady state constraints as proposed by [48]. To circumvent unidentifiability issues, several parameters (denoted with an asterisk in S1 and S2 Tables) were fixed prior to parameter estimation. The model was initialized 100 times with random parameter sets to find the system's local optima. To ensure sufficient variability between these initial parameter sets, we systematically sampled the parameter space with Latin hypercube sampling [49]. Best-fitting model parameter values are provided in S1 (default DDR model) and S2 (alternative DDR model) Tables. We tested the biological relevance of the fit by exploring the behavior of the model upon MDM2-p53 feedback disruption. We simulated this disruption by multiplying the $kd_{p53\ mdm2}$ and $kd_{p53p\ mdm2}$ parameters with 0.2 or 0.5 for respectively 20% or 50% feedback efficiency. From the 38 estimated parameter sets with minimal cost, each corresponding to this optimal fit, a random set was selected as a base for virtual sample generation.

**Virtual samples.** One thousand sets of 50 virtual samples per set were generated based on the best-fitting parameter set. To simulate these virtual samples, we considered variability across samples from different PHH donors to primarily originate from differences in the reaction rates within the underlying network, which could for instance be due to differences in expression levels of non-modeled proteins. First, we introduced variability in the value of parameters $p$ by adding a value $x$ to each parameter, where $x$ is drawn from a normal distribution $\mathcal{N}(\mu = 0, \sigma_p = c \cdot p)$, with $c \in \{0.001, 0.01, 0.1, 0.2\}$. Second, we analytically determined the steady states of all model species per modified parameter set with the fsolve function of the scipy package in Python version 3.7.2. Third, we simulated the response of the virtual PHH samples to cisplatin treatment by starting from steady state at $t = 0$ and applying a stress level $S$, after which we determined the expression of *TP53* and *MDM2*, *CDKN1A* and *BTG2* at the 8- and 24-hour timepoint (without applying the observable functions). The value for $S$ at nominal concentration 3.3 μM (as utilized in the experiments with PHHs) was estimated by linear interpolation. Furthermore, we examined the role of several reactions on the *TP53-MDM2* correlations by changing the strength of p53 dephosphorylation rate $k_{dp}$, MDM2 synthesis rate $ks_{mdm2\ p53p}$, or MDM2-p53 feedback $kd_{p53\ mdm2}$ and $kd_{p53p\ mdm2}$. To this end, we increased or decreased these parameters by multiplying them with factor $r$, where $r \in \{0.01, 0.1, 1, 10, 100\}$, and repeated the procedure to create 1000 virtual PHH donor sets per r-factor, while setting $c = 0.2$. The parameter sets that gave negative steady state values, which occurred in less than 0.75% of the sets, were removed. We introduced measurement noise $y'$ for each expression value $y$ in our model simulations by drawing $\ln(y')$ from a normal distribution $\mathcal{N}(\mu = \ln(y), \sigma = 0.125)$, which led to a good match between PHH donors and virtual donors with respect to the correlation for control versus cisplatin-treated *TP53* expression correlation at variation $c = 0.2$. We used one thousand bootstrapping replicates by randomly selecting 50 PHH donor samples with replacement to obtain the 95% confidence interval around the observed correlations.

## Results

### DNA damage-related gene activation upon cisplatin treatment is similar in HepG2 cells and PHHs

To obtain insight in the extent of variability among PHH donor samples and how the hepatocellular carcinoma cell line HepG2 relates to PHHs in terms of their response to DNA damage, we exposed both cell types to cisplatin at concentrations ranging from 0.1 to 100 μM. Cisplatin induced DNA damage in HepG2 cells after exposure, indicated by an increase in the number of γ-H2AX foci evident at 16 and 24h post exposure (S1A Fig). As expected, p53 became active

in a dose dependent manner, as evident from its phosphorylation at two distinct sites and the induced expression of p21 (S1B Fig).

We evaluated gene expression profiles by TempO-Seq analysis in combination with the targeted S1500+ gene set [36] for PHH samples derived from 50 donors and for HepG2 cells exposed to cisplatin (S1C Fig). Dimensionality reduction based on the normalized expression of all genes in the transcriptomics data set neither revealed outlying donors nor clearly distinguishable subgroups of donor samples, except for the HepG2 cells that had clearly different gene expression patterns (S1D Fig). We further explored the similarity in differential gene expression profiles of HepG2 cells and PHHs in response to cisplatin-induced DNA damage by comparing the activated genes based on PHH TXG-MAPr gene sets termed 'modules' [41]. Modules 83, 59 and 391, associated to p53 signaling and DNA damage, were among the twenty modules with the highest eigengene scores in HepG2 cells or PHHs exposed for 8 or 24 hours to 3.3 μM cisplatin (S1E Fig) and in the top 20 activated modules in response to the DNA damaging compound etoposide in the PHH TXG-MAPr. Thus, the cell types demonstrated a similar induction of the DDR, although there were also differences in other upregulated modules, even within cell types.

Following this observation of similar DDR induction in HepG2 cells and PHHs, we zoomed in on specific genes within the DDR. Specifically, we selected the genes *TP53*, *MDM2*, *CDKN1A* and *BTG2* (Fig 1A) for further analysis, because we had access to HepG2 reporter cell lines in which protein expression resulting from these genes could be measured over time. After normalization of the TempO-Seq data, we applied hierarchical clustering on *TP53* expression profiles for all concentrations and two time points. Based on this clustering, we divided the donor samples into three groups having low, intermediate or high *TP53* expression across all cisplatin concentrations (Fig 1B). The HepG2 cell line clustered with one PHH donor with lowest *TP53* expression levels. HepG2 and PHH sample clustering based on *MDM2*, *CDKN1A* or *BTG2* expression resulted in different clusters, where the HepG2 cell line clustered with PHH donors in the low (for *CDKN1A*), intermediate (for *BTG2*) or high (for *MDM2*) expression clusters (S2A–S2D Fig). Interestingly, the difference in *TP53* expression between the clusters was also observed when only considering control conditions, i.e., the basal expression (Fig 1C). In addition, HepG2 cells had considerably lower basal expressions than the average levels of the PHHs it clustered with for all genes except *MDM2* (Fig 1D–1F), whereas the expression after cisplatin exposure was generally higher in HepG2 cells than in PHHs (S2E–S2H Fig).

To further characterize the similarity between HepG2 cells and PHHs from an 'average' donor, we specifically investigated the dose-response relationship for cisplatin exposure to both cell types. *TP53* expression showed only a limited increase with cisplatin concentration in HepG2 cells (Figs 2A and S3A, left panel). In contrast, there was no concentration-dependency of *TP53* expression in PHHs when considering all donors together (Figs 2A and S3A, compare left and right panel). Compared to the weak concentration-dependency in *TP53*, the expression of downstream target genes *MDM2*, *CDKN1A* and *BTG2* in HepG2 cells and PHHs exhibited considerably stronger dose-response relationships (Figs 2B–2D and S3B–S3D, compare left and right panels). Although the $EC_{50}$ determined through dose-response curve fitting was somewhat variable among cell types and time points, the difference between cell types was typically less than 10-fold (S3E–S3H Fig). Note that there was substantial measurement noise, which was visible as weak correlations between technical replicates especially for *CDKN1A* (S3I–S3L Fig). This likely results from the relatively low expression of this gene, because we observed similarly weak correlations for the 10 lowest expressed genes (S3M Fig), but this finding does weaken a conclusion about the potential dose-response effect in the *CDKN1A* gene. Gene expression of all four DNA damage-related genes in both HepG2 cells and PHHs

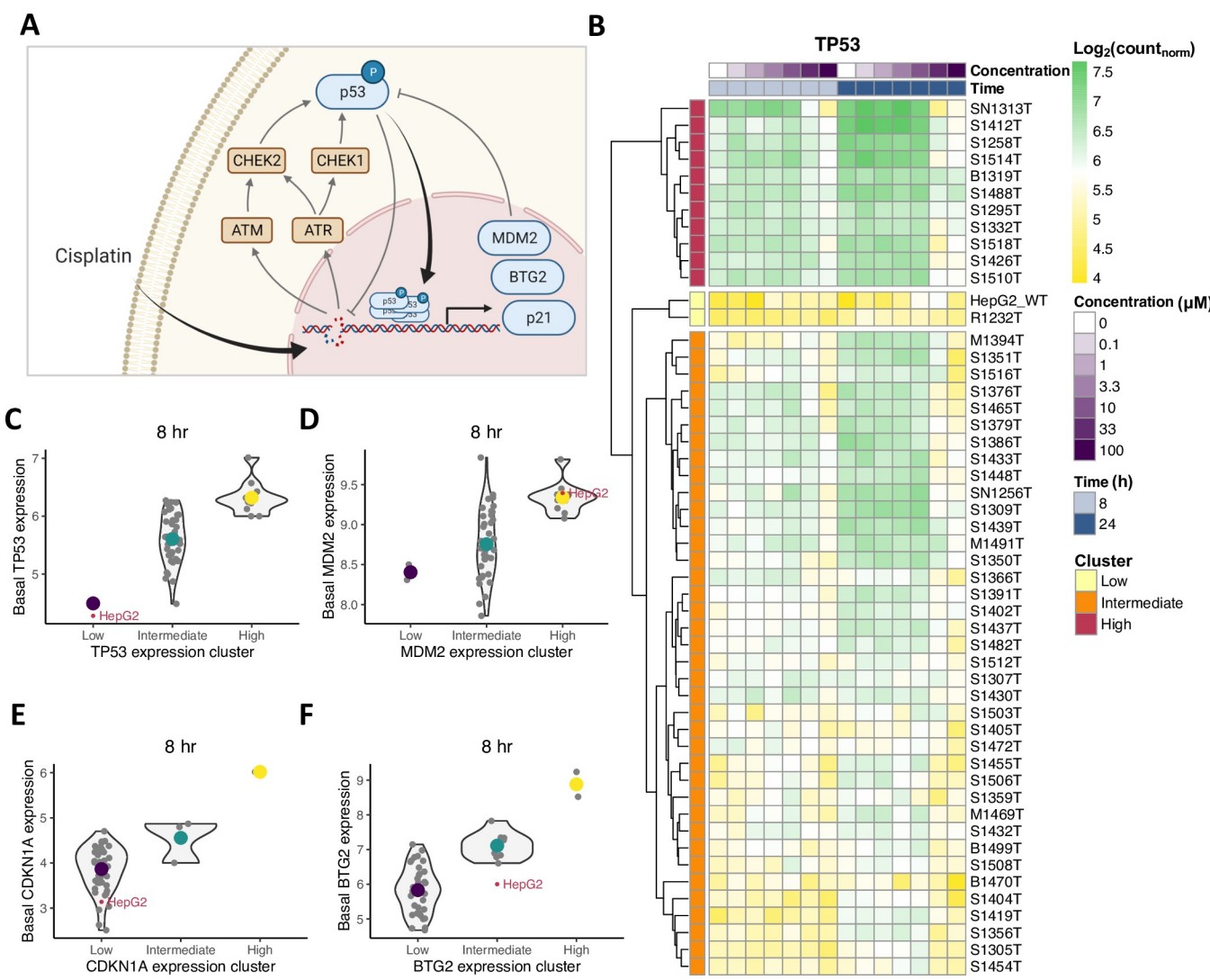

**Fig 1. Variability in basal and cisplatin-induced gene expression in HepG2 cells compared to PHHs from donor samples.** (A) Graphical representation of DNA damage-induced protein signaling upon cisplatin exposure. Kinases ATM, ATR, CHEK1 and CHEK2 are activated upon DNA damage and phosphorylate transcription factor p53. Phosphorylated p53 enters the nucleus and induces transcription of downstream targets, among which are MDM2, p21 and BTG2. (B) *TP53* expression in HepG2 and 50 PHH donor samples without and after cisplatin exposure. HepG2 cells cluster with one low *TP53*-expression PHH donor sample. (C-F) Basal expression of *TP53* (C), *MDM2* (D), *CDKN1A* (E) and *BTG2* (F) in PHHs and the HepG2 cell line within their corresponding low-, intermediate- and high-expression clusters. HepG2 cell line has lower basal expression levels for *TP53*, *CDKN1A* and *BTG2*, but not for *MDM2*, compared to average expression in PHHs within the same cluster. Contour lines are violin plots with individual samples marked by small grey dots, and large colored dots are cluster means.

decreased for cisplatin concentrations higher than 10 μM. Functional enrichment analysis indicated differential gene expression for cell death- and cell cycle-associated terms at concentrations of 10 μM and higher (S4A and S4B Fig). In addition, protective mechanisms such as positive regulation of p21 and negative regulation of apoptosis were among the highest enriched GO terms at 0.1 and 3.3 μM, whereas apoptosis-related GO terms were significantly enriched at 10 and 100 μM (S4C–S4G Fig). To prevent interference of cell death events with gene and protein expression, concentrations of 10 μM and higher were excluded from subsequent analysis. Taken together, the DNA damage related expression patterns of *TP53*, *MDM2*, *CDKN1A* and *BTG2* in HepG2 cells are similar to those in PHHs, although there are notable

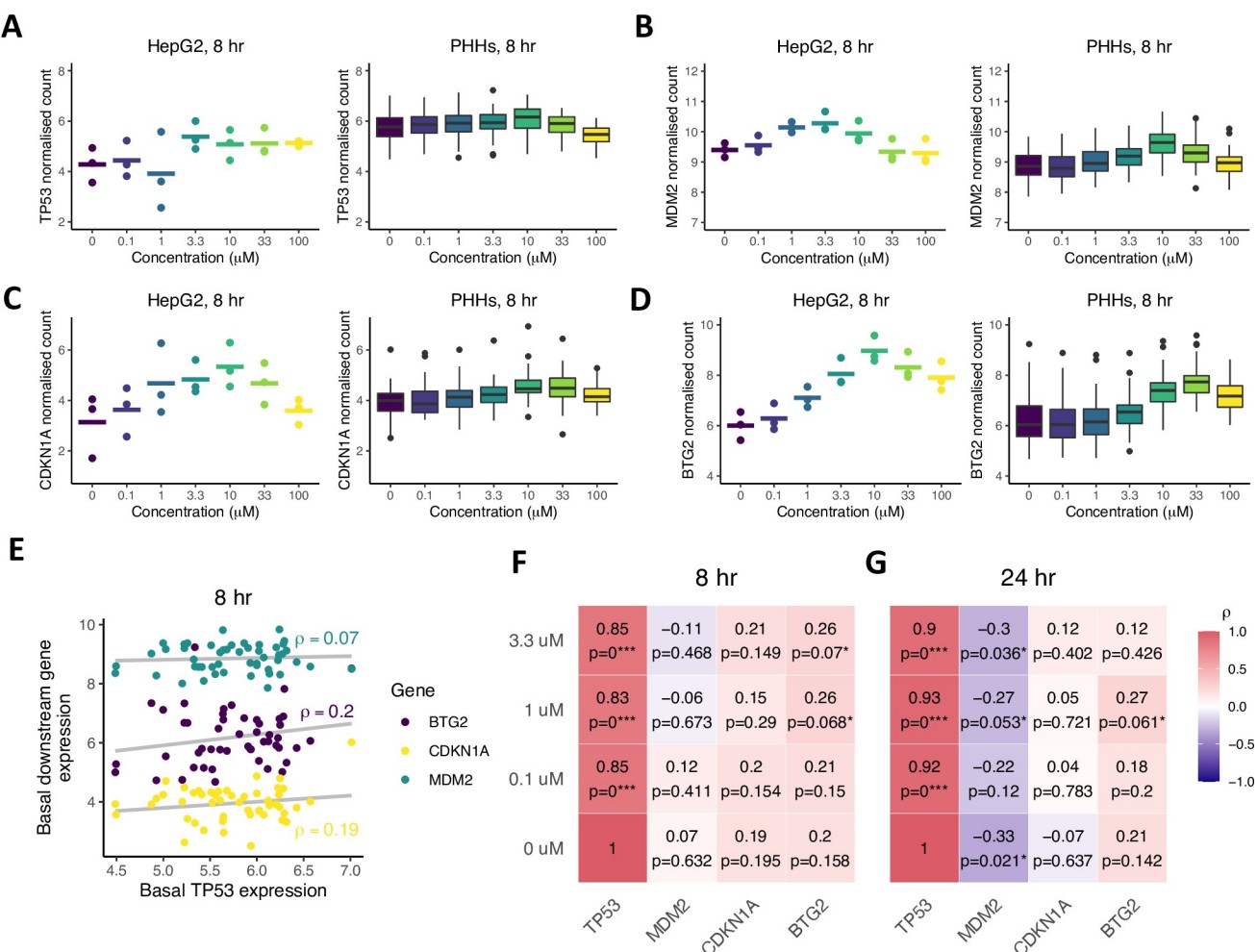

**Fig 2. Expression patterns and correlations between *TP53* and its downstream targets in HepG2 cells and 50 PHHs at increasing cisplatin levels.**
(A-D) *TP53* (A), *MDM2* (B), *CDKN1A* (C) and *BTG2* (D) gene expression patterns as a function of cisplatin concentration in HepG2 cells (3 replicates (dots) and their mean (line segments), left panels) and PHHs (right panels) at the 8-hour time point. Note that at high cisplatin concentrations, gene expression declines, which is likely explained by cytotoxicity onset. (E) Basal downstream target correlations with basal *TP53* expression at the 8-hour timepoint. (F-G) Correlation strengths between *TP53* and the downstream targets *MDM2*, *CDKN1A* and *BTG2* after exposure to 0, 0.1, 1 and 3.3 μM cisplatin at 8 (F) and 24 (G) hours. * p-value < 0.1, ** p-value < 0.01, *** p-value < 0.001.

differences in the relation between *TP53* expression and the expression of the downstream targets. Therefore, we continued to study the relations between *TP53* and its downstream targets.

## Basal *TP53* expression levels correlate weakly with downstream targets after cisplatin treatment in PHHs

Due to the extent of variability in expression of all four DNA damage associated genes among PHH donor samples, the question arose whether differences in donors' sensitivity to DNA damage could be predicted based on the *TP53* expression in a donor. To this end, we performed Pearson correlation analysis of basal *TP53* expression in PHHs and the three downstream targets before and after cisplatin exposure. Since the increase in p53 protein expression upon DNA damage is primarily caused by enhanced translation and posttranslational processes rather than increased transcription [50], p53 mRNA should not be greatly affected by increasing exposure to chemicals and should not radically change over time in control

conditions. Correlations between PHH basal *TP53* expression and the expression in PHHs exposed to cisplatin for 8 and 24 hours were indeed strong and there was no concentration-dependency (S5A Fig). Moreover, there was only a limited increase in *TP53* expression over time for PHHs with low *TP53* expression (S5B Fig). Next, we examined the relation between the basal expression of transcription factor *TP53* and the expression of the three *TP53*-regulated genes *MDM2*, *CDKN1A* and *BTG2*. In the non-treated conditions, basal *TP53* levels correlated weakly (albeit not significantly) with the basal expression of its downstream targets at the 8-hour timepoint (Fig 2E and 2F, bottom row). The correlations with basal *TP53* were slightly stronger after exposure to cisplatin, in particular for *BTG2* (Fig 2G). The correlations at the 24-hour timepoint were weaker than those at the 8-hour timepoint, except for the correlation between *TP53* and *MDM2* expression that became stronger (Fig 2G). Overall, correlations between *TP53* and target gene expression were low and barely significant. Interestingly, basal *TP53* expression negatively correlated with *MDM2*, both in absence and presence of cisplatin, although the strength of these negative correlations differed among the three *MDM2* probes used for sequencing (S5C Fig). Moreover, application of the same correlation analysis stratified by *TP53* expression group demonstrated that the strongest correlation occurred for the high-expression group (S5D Fig). In summary, our findings suggest a limited predictive capacity of basal *TP53* expression to PHH sensitivity for DNA damage.

## An ODE model describes p53 pathway activation upon cisplatin-induced DNA damage in HepG2 cells

Since availability of PHHs is limited and long-term usage of PHHs is compromised by their *in vitro* functional instability [10], hepatocellular cell lines such as HepG2 cells are often used as a biological model system instead. To translate between findings in HepG2 cells and PHHs, we asked whether the HepG2 cell line could be used to predict the response in PHH samples. To assess the applicability of HepG2-based predictions to PHHs, we aimed to generate virtual PHH donor samples derived from a deterministic ODE model that describes p53, MDM2, p21 and BTG2 mRNA and protein dynamics in HepG2 cells after cisplatin-induced DNA damage (Fig 3A). First, we utilized the protein expression data from Wijaya et al. (2021) obtained by measuring intracellular GFP intensity over time with live-cell microscopy of earlier generated HepG2 p53-GFP, MDM2-GFP, p21-GFP and BTG2-GFP reporter cell lines [18–20] (Fig 3B). This high throughput imaging method was used to measure protein levels up to 72 hours in hundreds of cells simultaneously on a single cell level, and to obtain population average protein expression over time (normalized data with background subtraction in Fig 3C, colored points; raw data in S6A Fig). We selected the data for conditions with 1, 2.5 and 5 μM cisplatin, since concentrations of 10 μM and higher caused decreased cell viability (S6B and S6C Fig), which is in correspondence to the functional enrichment analysis results for cell death in PHHs (S4 Fig). Upon cisplatin exposure, p53 levels peaked between 35 and 42 hours, depending on the cisplatin concentration. As expected, MDM2, p21 and BTG2 reached their maximum expression levels later than p53 (Fig 3D). In addition, we determined the initial response latency, defined as the time point of the first considerable increase in downstream target expression relative to this time point measured for the p53 protein. An increase in p53 protein expression could be distinguished around 5 hours after cisplatin treatment, whereas the initial response in MDM2, p21 and BTG2 proteins was delayed with at least two hours (Fig 3E).

Interestingly, the response to cisplatin-induced DNA damage consisted of sustained p53 expression, which is opposed to the more widely documented oscillatory behavior elicited by stressors such as radiation [51–55]. Therefore, we construct a new ODE model that could be used to simulate the time-resolved protein dynamics of p53, MDM2, p21 and BTG2 (Fig 3A,

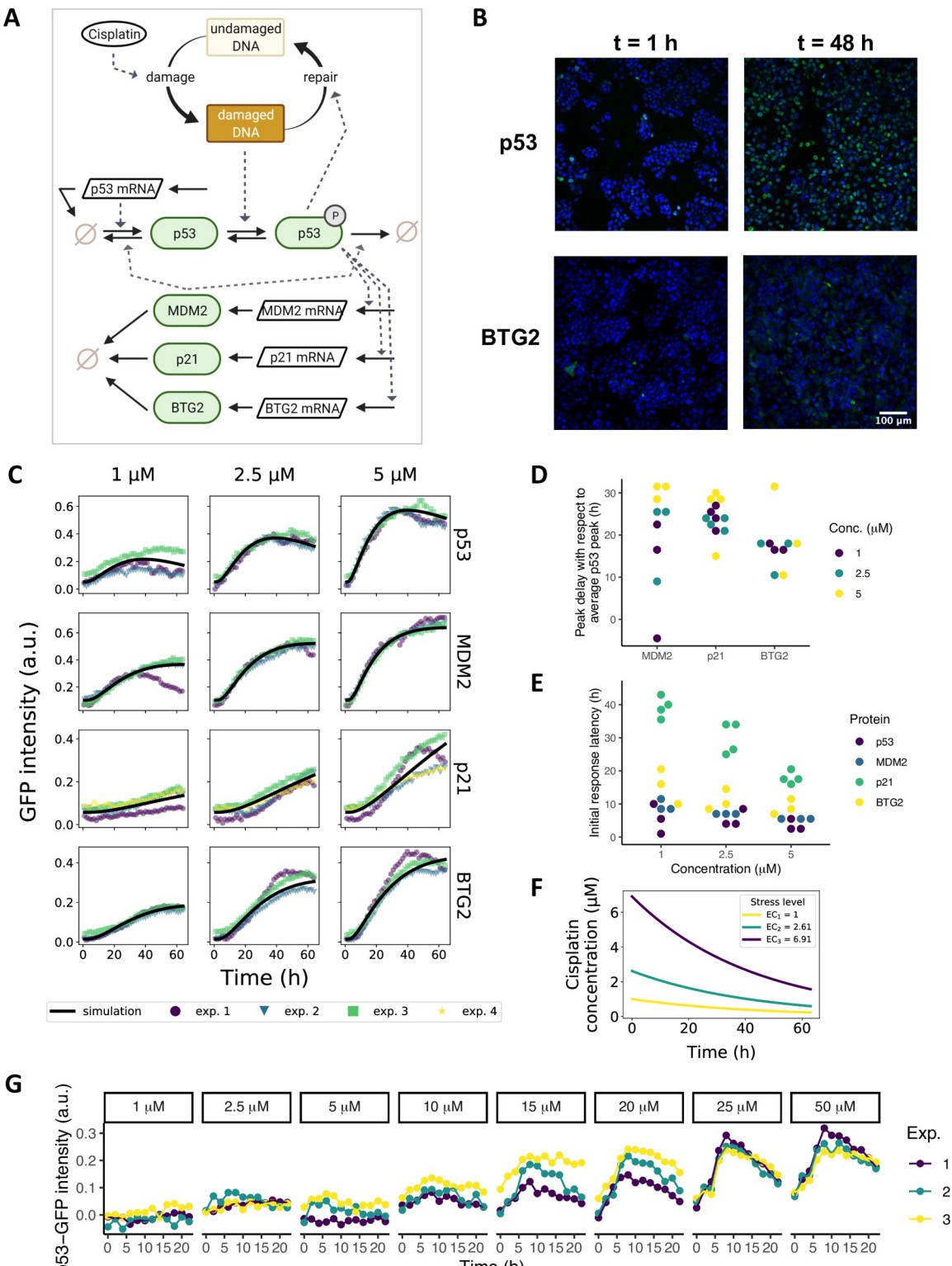

**Fig 3. ODE modeling of cisplatin-induced p53 signaling in HepG2 cells calibrated on live cell confocal imaging data.** (A) Graphical representation of the model. Cisplatin causes DNA damage, which triggers phosphorylation of p53. Phosphorylated p53 induces expression of its downstream targets MDM2, p21 and BTG2. (B) Example images of nuclear GFP expression in p53, and cytoplasmic GFP expression in BTG2 HepG2 BAC-GFP reporter cells at 1 and 48 hours. Blue, Hoechst-stained nuclei; green, GFP signal. (C) Mean of single-cell protein expression data measured over a 65-hour period after 1, 2.5 and 5 μM cisplatin exposure. Experimental replicates

(exp. 1–4) are shown in different colors. The model simulation after parameter calibration is shown as a black solid line. (D) Delay in peak MDM2, p21 and BTG2 expression levels with respect to maximum p53 expression. (E) Response latency of MDM2, p21 and BTG2 for different cisplatin concentrations. (F) Model-predicted effective cisplatin concentrations ($EC_i$) over time. Corresponding applied concentrations are 1, 2.5 and 5 μM for $EC_1$, $EC_2$ and $EC_3$, respectively. (G) Induction of p53-GFP in three replicates (Exp. 1–3) after exposure of non-treated HepG2 cells to conditioned medium collected from cells previously exposed to cisplatin for 72 hours.

Methods) and fit it to the imaging data (Fig 3C, black lines). To accurately describe effects on the protein dynamics, we included upstream cisplatin availability and the ensuing DNA damage stress in the model. We considered the reactive cisplatin concentration to decay exponentially (Fig 3F). To investigate whether cisplatin indeed disappears relatively quickly at low concentrations as predicted by the model fit, we indirectly determined the residual active cisplatin in medium after 72 hours of cell exposure by exposing fresh reporter cells to the supernatant. Consistent with model predictions, after 72 hours of 1, 2.5 and 5 μM cisplatin exposure, there was only a negligible amount of reactive cisplatin left in the medium, as evident from the absence of p53 response to the re-used medium. In contrast, residual reactive cisplatin for applied concentrations of 15 μM and higher still induced a stress response in p53 reporter cells following exposure of fresh reporter cells to supernatant, and this response increased with applied concentration (Fig 3G). Thus, our model fit correctly describes the relatively fast cisplatin decay at low concentrations.

Beyond the kinetics of cisplatin, the ODE model further includes cisplatin-induced DNA damage, which results in p53 activation by post-translational modifications such as phosphorylation (S1B Fig), acetylation and deubiquitination, causing cellular accumulation of p53-p [56–58]. In the model, the DNA repair rate depends on the amount of phosphorylated p53 because of its known regulatory role in nucleotide excision repair [26], which generates a negative feedback between p53-p and DNA damage. A second negative feedback between p53-p-dependent transcriptional activation of MDM2 and MDM2-induced degradation of p53 and p53-p was required for downregulation of the total amount of p53 at later time points. We added mRNA species for all four considered genes to the model, with the constraint that for p53 the protein accumulation does not rely on increased p53 mRNA production [50,59], i.e., the modeled p53 mRNA is considered to remain constant after exposure to cisplatin. In contrast, MDM2, p21 and BTG2 are primarily regulated on a transcriptional level, thus in the model the mRNA species for these genes change dynamically. The transcriptional activation of p21 and BTG2 by p53-p was modeled in a similar manner as for MDM2, using a Hill equation with exponent 4 to reflect p53-p binding to the DNA as a tetramer. Note that a model in which this is changed to a Hill exponent of 1 (neglecting p53-p tetramer formation) could also describe the protein expression data reasonably well, but predicted unrealistic behavior at late time points for MDM2, BTG2 and p21 (S7A and S7B Fig). To prevent overfitting, we did not include additional model components for which we have no experimental information.

Parameter optimization resulted in a fit that provided a good description of the GFP reporter imaging data (Fig 3C, black lines and S1 Table), with parameter uncertainty as shown in S8 Fig. To examine whether this optimal model parameterization correctly reflects the regulation of p53 degradation by MDM2 in HepG2 cells, we performed *in silico* disruptions of the MDM2 inhibitory effect on p53. A reduction of the MDM2-dependent p53 degradation had a clear effect on total p53 levels (Fig 4A). Next, we experimentally examined the effect of MDM2 on p53 degradation in HepG2 cells by inhibiting the MDM2-p53 interaction with Nutlin. The imaging data resulting from this experiment exhibited a profound suppressive effect of MDM2 on total p53 levels, which was evident from both the control case without cisplatin and the case with cisplatin (Fig 4B). The MDM2 response itself also increased strongly upon Nutlin treatment (Fig 4C), presumably because of the strong p53 induction. In conclusion, with the

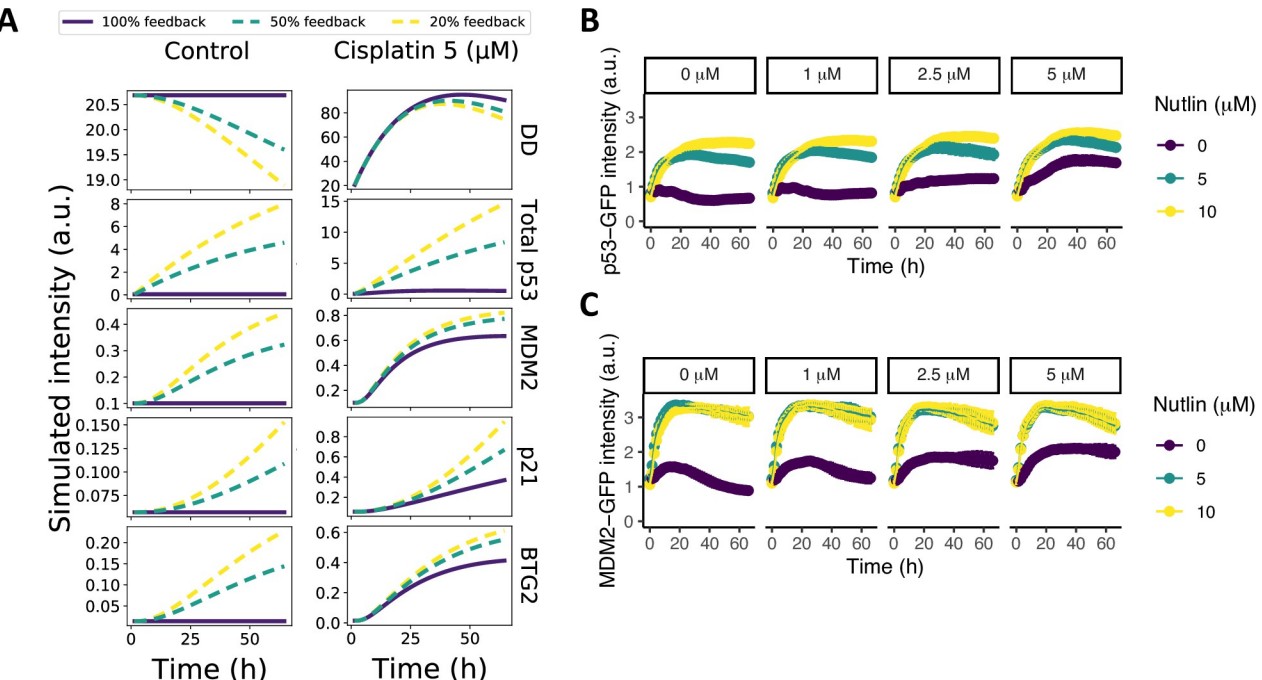

**Fig 4. Reduced p53-MDM2 feedback promotes p53 and MDM2 expression.** (A) Model simulations upon disruption of the MDM2-p53 feedback by weakening the feedback strength of MDM2 on p53 degradation to 50% (dashed green) or 20% (dashed yellow) of the original strength; simulations without disruption are shown as solid purple lines. (B-C) p53-GFP (B) and MDM2-GFP (C) expression in HepG2 cells after exposure to increasing concentrations of cisplatin (purple) and to cisplatin in combination with 5 (green) or 10 (yellow) μM Nutlin, both administered at t = 0 hours.

developed model we accurately describe the protein dynamics observed in HepG2 cells, while the inner model states corresponded qualitatively to the experimental data on cisplatin activity over time and the MDM2 negative feedback on p53 in our assays.

## Approximation of PHH with model-derived virtual donor samples

Having a mathematical model in place to describe the response of liver cells to cisplatin, we next examined the potential of our HepG2-based model in explaining the experimentally observed variability with respect to the studied DNA damage components in the 50 PHH donors. We therefore used the model to generate groups of 50 virtual donors that could be compared to the PHH expression data. Since the estimated parameter values were optimized to describe the protein dynamics in HepG2 cells, these values are specific for this system. Although reaction rates in PHHs are likely to differ from those in HepG2 cells and to be variable among PHH samples, the HepG2 based parameters may still form a good foundation to base expected parameter variability on. To simulate virtual donors (Fig 5A), we therefore introduced parameter variability proportional to the HepG2-based parameter values and ran simulations with each randomly chosen parameter set to determine the expected correlations between basal *TP53* levels and downstream target expression. Since we have no data on the actual parameter values underlying the variability between individual PHHs, we included a variability factor *c* that quantified the extent of differences between HepG2 and PHH samples, and we chose this from a range between 0.001 and 0.2, covering very little to high amounts of variability. Moreover, because measurement noise must also have affected the experimentally observed correlations, we added a small amount of such noise to the virtual donor data (see Methods).

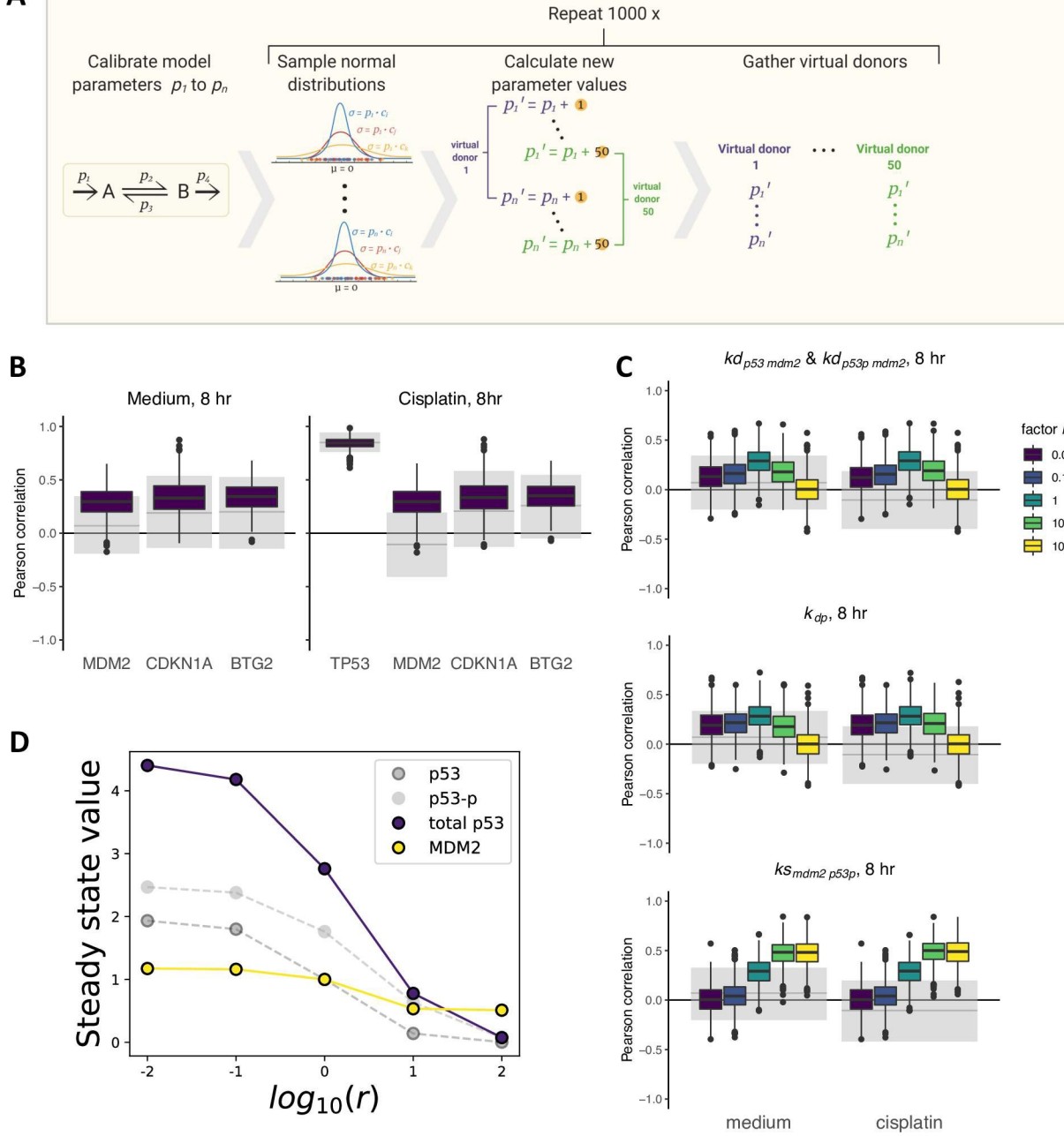

**Fig 5. Comparison between correlations for virtual donor samples and PHHs.** (A) Graphical explanation of the creation of virtual donor samples. For all fitted parameters $P_i$, we added a value, drawn from a normal distribution with mean 0 and standard deviation $p_i \cdot c$, to the parameter value. We executed this 50 times to generate one set of 50 virtual donor samples and repeated this 1000 times to obtain 1000 sets of 50 virtual donor samples. (B) Correlations between basal *TP53* expression and the downstream targets *MDM2*, *CDKN1A* and *BTG2* at 8 hours for HepG2-derived virtual samples (colored boxes) and PHHs (grey shaded areas) in basal expression conditions (left) and after 3.3 µM cisplatin exposure (right) with variability factor $c$ = 0.2 and a small amount of measurement noise, chosen such that the correlation of basal *TP53* with itself after cisplatin exposure under variability factor $c$ = 0.2 decreased to the experimentally observed correlations in PHHs. The grey solid lines represent the observed correlations in PHHs and the grey shaded areas represent the 95% confidence interval of correlations found for alternative PHH donor sets acquired with 1000 times bootstrapping. (C) Effect of changes in MDM2 feedback strength (top), dephosphorylation rate (middle) and p53-dependent MDM2 synthesis rate (bottom) on correlation of basal *TP53* expression with basal *MDM2* expression (medium) or after 3.3 µM cisplatin exposure (cisplatin) at the 8-hour timepoint and with measurement noise. A value r = 1 implies no adjustment in parameter value with respect to the fitted values, r > 1 implies stronger feedback and r < 1 implies weaker feedback. Grey solid lines and shaded areas are the same as in (B). (D) Adjustments in steady state values of p53 and MDM2 upon changes in factor r.

Utilizing these ODE model simulations of 50 virtual PHH donors (S9 Fig), we obtained the *TP53*, *MDM2*, *CDKN1A* and *BTG2* mRNA levels at 8 hours to determine the correlation between these components among a set of virtual donors. Furthermore, we simulated 1000 virtual donor groups to determine the expected variability in the correlations. We compared these *in silico* correlations to the *in vitro* correlations, considering the uncertainty in the experimental estimate by bootstrapping, i.e., rendering an estimate for what would be found in other sets of 50 PHH samples. For $c = 0.2$ and a small amount of measurement noise, the correlations between basal *TP53* and the downstream targets *CDKN1A* and *BTG2* within virtual patient groups closely matched those observed in the PHHs both without and after cisplatin treatment (Fig 5B). Note that without addition of measurement noise, the measured correlations in PHHs remained somewhat weaker than the median of the correlations in virtual HepG2-derived samples, even at high introduced parameter variability factor $c$ (S10A Fig).

In contrast to the close correspondence between measured and predicted correlations of *TP53* to *CDKN1A* and to *BTG2*, there was a mismatch for the *TP53-MDM2* correlations that was most profound in the condition with cisplatin exposure. Therefore, we investigated whether a change in parameters that influence p53 or MDM2 levels altered the correlation strength. For this purpose, we increased and decreased the MDM2-dependent p53 and p53-p degradation rate, the p53-p dephosphorylation rate and the p53-dependent MDM2 synthesis rate by multiplying these parameters with factor $r$ before introducing variability of this parameter with factor $c = 0.2$. Strong ($r = 10$, $r = 100$) or weak ($r = 0.01$, $r = 0.1$) MDM2 feedback strength led to a decrease in the *TP53-MDM2* correlation (Fig 5C, top panel and S10B Fig). Similarly, changes in the p53 dephosphorylation rate and p53-dependent MDM2 synthesis rate changed the correlation between *TP53* and *MDM2* (Fig 5C, middle and bottom panel, and S10C and S10D Fig), but did not lead to a negative correlation. In our model, the steady state expression levels of p53 and MDM2 simultaneously increased or decreased upon changes in MDM2 feedback strength, which explains the preservation of their positive correlation (Figs 5D and 4B and 4C). We investigated two additional model changes for their ability to explain negative correlations between *TP53* and *MDM2* across PHH donors. The first change involved implementation of a non-linear rather than linear MDM2 feedback on p53, but this did not lead to a negative correlation (S11A Fig). The second change consisted of an alternative model, in which MDM2 is phosphorylated and binds to p53 mRNA to induce p53 translation [60,61] (S11B and S11C Fig). Also this model could not explain the negative *TP53-MDM2* correlation (S11D Fig). In summary, although we found good correspondence between correlations of *TP53* with its downstream targets *CDKN1A* and *BTG2* in PHHs and HepG2-derived virtual donor samples, our model could not explain the negative correlation between *TP53* and *MDM2* observed for PHHs.

## Discussion

There is a need to understand the overall relevance and uncertainty of cell line usage as model system for healthy human tissue, which requires detailed understanding of how intracellular network dynamics translate between cell types from different genetic backgrounds. Here, we used correlations between protein and gene expression levels to infer the similarity of DNA damage response dynamics in the HepG2 cell line with the response of PHHs. We showed that our mechanistic ODE model calibrated on HepG2 protein expression data could reproduce the relation between the expression of *TP53* and its downstream targets *CDKN1A* and *BTG2* in PHHs, but not the negative correlation between *TP53* and *MDM2*. Our method allowed us to explore the applicability of a HepG2-derived model to explain heterogeneity in PHHs and identify knowledge gaps in pathway regulation. Ultimately, this approach can be used to

provide new insights in the applicability of results obtained from routine, high throughput cell line test systems for the responses of primary human cells.

For reliable extrapolation of the DNA damage response dynamics in HepG2 cells to PHHs, the genetic differences among liver tissue samples should be considered. Firstly, various studies reveal cisplatin-induced DNA damage in PHHs and HepG2 cells, but the extent of damage varies between these cell types, which may in part be due to differences in assay sensitivity among cell types [62–64]. Secondly, the extent of DNA damage [63] and consequent gene expression varies among PHH donors (S2 Fig), which is in line with previous findings [65,66] and more extensively studied in the here utilized data of Niemeijer et al. (2021).

Furthermore, the HepG2 samples were clearly separated from the PHH samples based on their gene expression profile, although HepG2 will likely be much more similar to PHHs compared to unrelated reference cell lines. The function of p53 in cell cycle regulation and apoptosis complicates the comparison of the proliferating HepG2 cell line with quiescent PHHs. We found a moderate temporal effect on *TP53* expression in PHHs, but not in HepG2 (Figs 1B and S5B), which could be due to the necessity of relatively high p53 expression in PHHs to maintain their quiescent phenotype [67]. There is indeed considerable genetic dissimilarity between unexposed PHHs and HepG2 cells, although this difference is less pronounced after cisplatin exposure [68]. In our analysis, hierarchical clustering based on individual gene expression patterns showed that HepG2 cells respond somewhat differently than any other PHH donor sample in this panel. For example, the HepG2 cell line clustered with PHH donor samples having low *TP53* expression, but clustering based on *MDM2* expression grouped the HepG2 cell line with PHHs that had high *MDM2* expression (Figs 1B and S2A–S2D). Indeed, overexpression of *MDM2* has been previously found in many cancer types including hepatocellular carcinoma [69–71], and MDM2 protein levels are also elevated in HepG2 cells [72]. Since HepG2 clustered with the PHH donor sample with lowest *TP53* expression, HepG2 cells could very well behave similar to PHHs with low basal *TP53* expression. If possible, extrapolation of simulated dynamics to PHHs should therefore be adjusted based on PHH characteristics. For experimental purposes, forced overexpression of the *TP53* gene in HepG2 cells might improve the resemblance to 'average' PHHs. Although gene expression was somewhat different between HepG2 cells and PHHs, we further investigated whether their pathway dynamics were nevertheless similar.

To this end, we sought to compare the dynamics of the p53 signaling pathway in these cell types by studying the correlations between *TP53* and its downstream targets at two time points. These correlations were rather weak, yet we found that similarly weak correlation strengths would be expected based on the virtual samples generated with our ODE model. Consistent with these findings, Spearman correlation coefficients ranging from 0.2 to 0.4 between p53, p21 and MDM2 protein expression have been reported for hepatocellular carcinoma tissue and surrounding hepatocytes [73]. The comparison of correlations in HepG2 virtual donors and PHHs suggested that our model captures the essence of the relationship between the expression of gene *TP53* and its downstream targets *CDKN1A* and *BTG2*. However, we found that adjusting the parameters that determine p53 or MDM2 expression was not sufficient to deliver HepG2-based virtual samples that reproduced the negative *TP53-MDM2* correlation observed for PHH donor samples. An alternative model that incorporates dynamics of *TP53* expression and a positive feedback of MDM2 on p53 production, could neither reproduce this negative correlation. Therefore, time-resolved mRNA expression data for *TP53* and *MDM2* to constrain their dynamics will be required to further elucidate the relation between p53 and MDM2 on transcriptional and translational level in HepG2 cells, and how this might differ from PHHs. Considering the architecture of our models and our findings that parameter manipulations were not sufficient to obtain a negative *TP53-MDM2* correlation,

additional factors that influence the dependency of MDM2 mRNA on p53 are likely also needed to accurately capture their relation. The MDM2-homologue MDMX, that modulates the p53-MDM2 autoregulatory feedback by enhancing MDM2-dependent p53 repression and inhibiting the transcriptional activity of p53 [74,75], could be a candidate for subsequent studies. Acquisition of mRNA and protein expression data of MDMX in HepG2 cells and PHHs, which is not available in the here utilized data sets, combined with inclusion of MDMX in our model might improve our understanding of the mechanisms that determines the MDM2-p53 relation and thereby the translation of the MDM2 protein dynamics from HepG2 to PHHs.

Former comparisons between *in vitro* biological test systems and PHHs have focused on highlighting similarities and differences in gene expression patterns. In this context, genes involved in metabolism have often gained special attention, since a better understanding of differences in pharmacokinetics between cell types can be used to interpret differences in genotoxic responses [76–78]. Here, we provided a method to investigate similarities between DNA damage response pathway activity among cell types on a mechanistic level, aiming to ultimately infer DDR pathway dynamics in PHHs. We showed that relations between mRNA expression in model-based virtual samples are comparable to these relations in PHH samples. Although the presented ODE model cannot yet fully explain the *TP53-MDM2* correlation within PHHs, our study has uncovered the MDM2-p53 feedback as critical factor for the translation of p53 pathway dynamics between cell types.

It is currently unclear what DDR-related protein expression levels are associated with good prognostic outcome e.g., for cancer patients treated with chemotherapy. For example, we do not know whether a high cisplatin-induced p21 expression is only associated with a good prognostic outcome for a patient, because high p21 expression is likely also associated with an increased probability for adverse effects in liver or kidney. Ultimately, advancements in our understanding on the relation between gene and protein expression dynamics and cell fate, will move us towards understanding the origin of interindividual differences in susceptibility to cancer development and will provide opportunities for the creation of patient-specific risk profiles.

## Supporting information

**S1 Fig. Response of HepG2 cells and PHHs to cisplatin-induced DNA damage stress.** (A) Mean number of γ-H2AX foci in HepG2 cells over time after cisplatin exposure (n = 1). (B) Expression of p53, p53-S15, p53-S46 and p21 at 24 (left) or 48 (right) hours after cisplatin exposure, as measured by Western blot (n = 1). (C) TempO-Seq library sizes per measurement and sample. Conditions with less than 100,000 reads (solid black line) are discarded from the analysis. CDDP, cisplatin. (D) Distribution of samples based on the expression of all genes in the S1500+ gene set after dimensionality reduction with PCA followed by t-SNE. (E) Overlap in the 20 TXG-MAPr modules with the highest Eigengene score per cell type at 8 and 24 hours after exposure to 3.3 μM cisplatin.
(TIF)

**S2 Fig. Variability in basal and cisplatin-induced gene expression in HepG2 cells compared to PHHs from donor samples.** (A-C) Gene expression heatmaps for *MDM2* (A), *CDKN1A* (B) and *BTG2* (C). (D) Gene expression cluster assignment for HepG2 cells and PHH donor samples based on hierarchical clustering. Note that clusters are not conserved among genes. (E-H) Basal expression of *TP53* (E), *MDM2* (F), *CDKN1A* (G) and *BTG2* (H) in PHHs and the HepG2 cell line within their corresponding low-, intermediate- and high-expression clusters. Contour lines are violin plots with individual samples marked by small grey dots, and means per cluster by large colored dots.
(TIF)

**S3 Fig. Expression patterns of *TP53* and its downstream targets in HepG2 cells and 50 PHHs at increasing cisplatin levels.** (A-D) *TP53* (A), *MDM2* (B), *CDKN1A* (C) and *BTG2* (D) gene expression patterns as a function of cisplatin concentration in HepG2 cells (three replicates (dots) and their mean (line segments), left panels) and PHHs (right panels) at the 24-hour time point. Note that at high cisplatin concentrations, gene expression declines, which is likely explained by cytotoxicity onset. (E-G) Dose-response curve fits based on a Hill equation for *MDM2* (E), *CDKN1A* (F) and *BTG2* (G), with data means per condition (empty colored markers) and $EC_{50}$ values (solid colored dots). (H) Fold change differences between $EC_{50}$ values per gene. (I-L) Correlation plots between technical replicates of PHH measurements for *TP53* (I), *MDM2* (J), *CDKN1A* (K) and *BTG2* (L). (M) Correlation coefficient between technical replicates (mean ± sd) of the 10 lowest (purple) and 10 highest (yellow) expressed genes.
(TIF)

**S4 Fig. Functional enrichment analysis for cytotoxicity related Gene Ontology (GO) terms based on transcriptomics measurements in PHH samples.** (A-B) Enrichment of selected GO terms for cell death (A) and cell health (B) terms at increasing cisplatin concentrations at the 8-hour time point. (C-G) The 10 most significant enrichment terms at 0.1 (C), 3.3 (D), 10 (E), 33 (F) and 100 (G) μM cisplatin. Note that the 1 μM condition is not included, due to the limited number of genes that passed the criteria for differentially expressed genes and the consequent absence of functional enrichment.
(TIF)

**S5 Fig. Gene expression correlations amongst PHHs from different donors.** (A) Correlations of basal *TP53* with *TP53* expression after 0.1, 1 and 3.3 μM cisplatin exposure at 8 (left) and 24 (right) hours. (B) Correlation between *TP53* expression at 8- and 24-hour timepoint. Dashed grey lines in (A-B), y = x. (C) Correlations of basal *TP53* with the three probes of *MDM2* (p1, p2 and p3) at 8 (left) and 24 (right) hours. (D) Correlations of basal *TP53* with *MDM2* (mean of the three probes) at 8 (left) and 24 (right) hours split per *TP53*-expression cluster. Grey dashed lines represent the overall correlation for all clusters together.
(TIF)

**S6 Fig. GFP, PI and AnV readouts of live-cell imaging data.** (A) Unnormalised GFP intensities for proteins and biological replicates separately. (B-C) Maximal proportion of PI-positive (B) and AnV-positive (C) cells at increasing cisplatin concentrations. Mean ± sd of all 13 experiments with the 3 to 4 replicates per BAC-GFP reporter.
(TIF)

**S7 Fig. Model dynamics at late time points differs due to a different Hill coefficient.** (A-B) Dynamics of the model species for a model with p53-dependent activation of transcriptional regulation with Hill parameter 4 (A) or 1 (B). DD, DNA damage.
(TIF)

**S8 Fig. Illustration of parameter estimate uncertainties across parameter calibration runs with the same cost function value.** A) Estimates of the ODE model parameters and initial states on a $\log_{10}$ scale. B) Estimates of the scaling and offset parameters used in the observable function on a linear scale.
(TIF)

**S9 Fig. Examples of expression pattern simulations in virtual donor samples.**
(TIF)

**S10 Fig. Impact of parameter manipulations on correlation strengths without measurement noise.** (A) Correlations between basal *TP53* expression and the downstream targets *MDM2*, *CDKN1A* and *BTG2* at 8 hours for HepG2-derived virtual samples (colored boxes) and PHHs (grey shaded areas) in basal expression conditions (left) and after 3.3 μM cisplatin exposure (right). Increasing variability in parameter values, i.e., increasing factor c, improves the match between the correlations in the HepG2-derived virtual samples and the PHH donor samples. (B-D) Effect of changes in MDM2 feedback strength (B), dephosphorylation rate (C) and p53-dependent *MDM2* synthesis rate (D) on correlation between *TP53* and *MDM2* after 8 (left) and 24 (right) hours. Correlations are shown for basal *TP53* expression with *MDM2* expression in medium or after 3.3 μM cisplatin exposure at varying parameter multiplication factors $r$. A value $r = 1$ implies no adjustment of the parameter value with respect to the fitted values, $r > 1$ implies stronger feedback and $r < 1$ implies weaker feedback. Colored boxes are the correlations found for HepG2-derived virtual samples. Grey horizontal lines represent the measured correlation for the 50 PHH donor samples. Grey shaded areas represent the 95% confidence interval of the correlation measurements acquired with 1000 times bootstrapping.
(TIF)

**S11 Fig. Correlation analysis with alternative DDR models.** (A) Correlations between basal *TP53* expression and the downstream targets *MDM2*, *CDKN1A* and *BTG2* at 8 hours for HepG2-derived virtual samples (colored boxes), created with a model with a non-linear p53-MDM2 feedback. (B) Alternative model in which p53 mRNA binds to phosphorylated MDM2 and induces expression of p53 mRNA. (C) Simulation of the alternative model (as in B) after parameter estimation. (D) Correlations between basal *TP53* expression and the downstream targets *MDM2*, *CDKN1A* and *BTG2* at 8 hours for HepG2-derived virtual samples (colored boxes), created with the alternative model (as in B). Correlations in B and D are shown for basal expression conditions (left) and after 3.3 μM cisplatin exposure (right) with variability factor c = 0.1 and a small amount of measurement noise for B, and variability factor c = 0.2 without measurement noise for D. Moreover, in B and D the grey solid lines represent the observed correlations in PHHs and the grey shaded areas represent the 95% confidence interval of correlations found for alternative PHH donor sets acquired with 1000 times bootstrapping.
(TIF)

**S1 Table. Parameter description and estimated values of the model as described in the Methods section of the main text.** Fixed parameter values are indicated with a diamond (⋄) and parameters that are determined with steady state constraint calculations with a star (⋆).
(DOCX)

**S2 Table. Parameter description and estimated values for the alternative DDR model described in S1 Methods.** Fixed parameter values are indicated with a diamond (⋄) and parameters that are determined with steady state constraint calculations with a star (⋆).
(DOCX)

**S1 Methods. Methods for the model with a non-linear p53-MDM2 feedback and the alternative DDR model.**
(DOCX)

## Acknowledgments

The authors gratefully acknowledge the Leiden University Cell Observatory imaging core facility for their support and assistance in this work.

## Author Contributions

**Conceptualization:** Muriel M. Heldring, Joost B. Beltman.

**Data curation:** Muriel M. Heldring.

**Formal analysis:** Muriel M. Heldring, Marije Niemeijer.

**Funding acquisition:** Bob van de Water, Joost B. Beltman.

**Investigation:** Muriel M. Heldring, Lukas S. Wijaya, Marije Niemeijer, Talel Lakhal.

**Methodology:** Muriel M. Heldring.

**Resources:** Sylvia E. Le Dévédec.

**Software:** Muriel M. Heldring, Huan Yang.

**Supervision:** Bob van de Water, Joost B. Beltman.

**Validation:** Muriel M. Heldring.

**Visualization:** Muriel M. Heldring.

**Writing – original draft:** Muriel M. Heldring.

**Writing – review & editing:** Lukas S. Wijaya, Marije Niemeijer, Huan Yang, Talel Lakhal, Sylvia E. Le Dévédec, Bob van de Water, Joost B. Beltman.

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
