## [Decision Letter · Decision Letter 0]

9 Feb 2022

Dear Dr. Beltman,

Thank you very much for submitting your manuscript "Model-based translation of DNA damage signaling dynamics across cell types" for consideration at PLOS Computational Biology.

As with all papers reviewed by the journal, your manuscript was reviewed by members of the editorial board and by several independent reviewers. In light of the reviews (below this email), we would like to invite the resubmission of a significantly-revised version that takes into account the reviewers' comments.

We cannot make any decision about publication until we have seen the revised manuscript and your response to the reviewers' comments. Your revised manuscript is also likely to be sent to reviewers for further evaluation.

Sincerely,

Pedro Mendes, PhD

Associate Editor

PLOS Computational Biology

Douglas Lauffenburger

Deputy Editor

PLOS Computational Biology

Reviewer's Responses to Questions

**Comments to the Authors:**

Reviewer #1: The authors create a mathematical model to predict dynamics of p53 mRNA and protein levels as well as protein levels of select p53 target genes in response to cisplatin induced DNA damage. Based on experiments conducted in HepG2 cells and primary human hepatocytes (PHH) the authors created a set of virtual PHHs to test this model. Thereby, the moderate correlations that the authors found in 50 biological PHH donor samples for p53, p21, and BTG2, was resembled in the virtual samples. The negative correlation between p53 and Mdm2 in the PHH donor samples could not be replicated with the model however. The authors conclude that HepG2-based computational modelling can be accurate for some, but not all DDR elements.

To frame our comments, we point out that we are molecular biologists, focussing on in vitro and in vivo investigations of the p53-Mdm2 feedback mechanism rather than computational/mathematical biologists. Therefore, we comment mainly on the biological context of the experiments and their findings. In general, the paper is well written and at each necessary point, the authors commit to its shortcomings and limitations, including its biological context. As such, the manuscript is well reflected and discussed with the context of currently available p53-Mdm2 literature (in HepG2 cells). We are aware that computational modelling of real-world problems is a challenging task and it comes along with simplifications. And we also believe that meeting these challenges is a valuable starting point for future developments.

Still, from the molecular biological point of view we want to raise the following concerns in response to the manuscript:

Major concerns

1. The authors do not provide direct evidence for DNA damage. Given the large disparity in gene expression in the response to cisplatin treatment in HepG2 and PHH cells, raises the question if DNA damage is inducible to equal extent in both systems. Comparing immortalized cell lines with timely limited culturable primary cells could lead to DNA damage-independent, but still p53-mediated, cellular responses. Therefore, we suggest showing direct evidence for DNA damage in both systems with the same concentrations as used in the manuscript i.e., gH2Ax induction, comet assay, 8OH-immunofluorescence or if available to provide a reference showing cisplatin-induced DNA damage in HepG2 cells or direct experimental evidence.

2. Although addressed as shortcoming in the text, we are concerned for the predictive capacity for CDKN1A, as the technical variability (Fig S3E-H) in the TempO-Seq analysis is quite high. How do the authors argue to utilize the data e.g., for CDKN1A with a Pearson correlation between 0.23 and 0.32, for a predictive model?

3. The conclusion of chapter one needs rework given the graphs shown in Figure 2. an objective measure (statistical value) for describing the data in Figure 2 (expression patterns of TP53, MDM2, CDKN1A and BTG2 in HepG2 compared to PHHs) should be provided, rather than merely stating that patterns are “similar”.

4. The experimental conditions change from 8h and 24h to 48h and beyond. Still the derived model predicts dynamics starting at timepoint zero, which was only experimentally obtained for mRNA data. We are inconclusive if such fundamentally different biological entities (mRNA, protein) experimentally measured at different timepoints should be integrated into one model. The authors are aware of pulsatile p53 mRNA regulation (or mRNA regulation in general as referenced in the text) and concluded in the first chapters, that PHH p53 mRNA levels were non-predictive. We ask to clarify how the TempO-Seq data is reflected in the equations.

5. The authors extend their experimental panel with co treatment of GFP cells with nutlin in an attempt to obtain data for p53-Mdm2 protein dynamics. Unfortunately, the timescale of the model (0-60h) does not match the provided data in Figure 5C (30-60h). The most dynamic phase of protein regulation, according to the provided model fits (0-30h) is therefore not covered but would potentially provide valid insight into this dynamic phase. Data for nutlin treatment was provided for later timepoints, when the protein abundances in the model is already plateauing. Please include the data from the time series for time points 0-30h.

6. The authors state in their discussion, p53 as well as Mdm2 mRNA levels have been shown to affect the protein levels of their respective counterpart. Therefore, we suggest adapting the model to the context of the well-known p53-mdm2 interaction, as we are confident that the body of p53-mdm2 literature potentially contains valuable additional information that could be included in the model. Potentially, this could strengthen the model in its predictive capacity. In fact, Mdm2 protein was previously described to directly affect p53 mRNA levels (DOI: 10.1038/ncb1770; DOI 10.1016/j.ccr.2011.11.016) which adds another layer of complexity on the p53-mdm2 regulatory mechanism. We suggest including these observations into the underlying equations in the model to better reflect known regulatory mechanisms. We therefore ask the authors again to consider adapting the model to include Mdm2 feedback into the p53 mRNA equation.

Minor concerns

1. Although the authors address the circumstance that p53 mRNA levels changes within the control conditions between 8h and 24h, we see this point critical. Fig 1B clearly indicates a time-dependent increase in p53 mRNA levels, which may be explained with difficulties in culturing PHHs and associated p53-mediated cell death (apoptosis) over time. How does a proliferating cell line (HepG2) and a quiescent state (PHHs) relate to each other? The authors partly addressed this difference with citing a paper in chapter 3, dealing with PHH stability in culture. However, p53 levels were not analysed in the cited paper. Given that p53 is involved in cell cycle regulation as well as cell death, we assume this reference as insufficient to explain p53 mRNA levels in PHHs. In consequence, we ask to authors to discuss this circumstance in more detail with additional literature.

2. The authors state that the p53 protein levels peaked between 35 hours and 42 hours. Anyhow, they do not show data for untreated controls only for cisplatin concentration 1, 2.5, 5µM. Showing untreated samples would answer the question if there are basal changes in DNA damage with culturing time. This also leads to the question whether the confocal microscopy data (GFP-protein) is normalized to nuclei counts?

3. The transcriptional activation of p21 and BTG2 by phosphorylated p53 was adapted to Mdm2 responsiveness in the model, which raises the question if this is a valid assumption, as the named targets react quite differently in the TempO-Seq analysis and experimental data about p53 phosphorylation is missing. We ask the authors to provide transcriptional activation data for p21 and BTG2 in either experimental setups or data from literature.

4. We observe that, the initial mRNA expression analysis and the later described GFP intensities, i.e., protein levels are not very well linked together. We ask the authors to comment on whether they normalize protein GFP signals to the respective mRNA levels and how experiments from TempO-Seq and GFP-protein overexpression are represented in their model, besides setting p53 mRNA levels to 1.

5. Figure 5A shows the two best fits of parameters for the model that the authors found. However, we fail to comprehend why p53 protein levels decrease upon DNA damage, while phosphorylation of p53 and protein levels of all target genes increase. Given p53’s role in DNA damage response, we would expect p53 protein levels to rise. Also, does the increase in p53-p take the decrease of total p53 into account in this model? Can this be normalized on total p53?

6. The y-axis limits are quite different in the graphs describing the model fits (Fig 5) and nutlin experiments. E.g., in Fig 5C, D the y-axis should start at zero, otherwise differences appear greater than they actually are and consequently mislead the reader.

7. Figure 6B shows that only the correlations of basal TP53 expression vs p53 protein levels after cisplatin treatment lines up between biological PHH samples and virtual samples. We ask how the authors ensure that their model is neither overfit nor underfit and discuss this accordingly.

Reviewer #2: Heldring et al., use a mixture of in vitro experiments and in silico modeling to study how a well described feedback loop governing p53 regulation. They take this data and combine it with medium scale gene expression data from PHH samples. Though the authors have both RNA and protein datasets the lack of protein data for the PHH donors ad complexity to their analysis. Overall, this paper both meets a critical need to develop approaches to leverage in vitro data to inform patient models and is insufficient to the task. As it stands the paper is only a weak fit for PLOS comp. Bio., however, plausible improvements might strengthen it considerably.

Major comments

(1) The computational model is thoughtfully designed, and well explained. Given the dominant role of transcription in regulation of downstream p53 targets, I would be interested to see if including mRNA species of p21/mdm2/btg2 would enable improved fits (obviously at a cost of higher complexity). In general, in addition to examining parameter sensitivity it would be useful if the authors looked at the robustness of their model to structural variation.

(2) The paper ends with a modeling approach to reconcile their patient data with a model of p53/mdm2 activity in response to cisplatin. Their virtual patient simulation is a reasonable approach. It would be very valuable to extend this analysis to its logical conclusion in giving treatment recommendations. If, for example, we assume high p21 levels are a ‘good’ clinical outcome what can we say about cisplatin dosing and the model parameters (and their steady state protein concentrations) that could achieve this outcome.

(3) In figure 1, how can you hierarchically cluster based on one number (p53 or mdm2 expression)? Do the authors mean they arbitrarily grouped the cell samples? This seems essentially descriptive.

(4) Looking at 4 genes from a 1500 gene set to determine ‘similarity’ seems very odd (even if they are important genes). The authors should use a larger set (either the whole 1500) or subsets drawn from published ‘gene-sets’.

(5) In figure 3 the authors seem to say they use p53 expression as proxy of cell survival/toxicity (341-343). As the authors later point out, this is a mis-reading of the literature. Instead, the authors could use the expression of cell cycle gene expression (eg ccnd1, aurora A, mki67) as one proxy of cell health (although this may be complicated in the phh cells), or perhaps the activation of NFKB targets (triggered by cytoplasmic accumulation of DNA) such as TNFAIP3. More generally the authors should use their expression data more extensively to provide a more complex picture of the overall response of the phh cells to cisplatin treatment.

Minor comments

(1) The paper figures are well drawn, but could be compacted considerably, the core point of the paper draws very little on data from figures 1-3 (for example figure 3 is basically a negative result) and this part could be condensed.

(2) Overall in the paper text there is some confusion between RNA and protein measurements and what each imply (not on the authors part I believe, but given their different measurements its important to be very specific in the text to avoid mis-reading). This is critical to this circuit so the authors should clarify this in each section.

**Have the authors made all data and (if applicable) computational code underlying the findings in their manuscript fully available?**

Reviewer #1: Yes

Reviewer #2: Yes

PLOS authors have the option to publish the peer review history of their article (what does this mean?). If published, this will include your full peer review and any attached files.

Reviewer #1: No

Reviewer #2: No
---

## [Decision Letter · Decision Letter 1]

30 May 2022

Dear Dr. Beltman,

We are pleased to inform you that your manuscript 'Model-based translation of DNA damage signaling dynamics across cell types' has been provisionally accepted for publication in PLOS Computational Biology.

Best regards,

Pedro Mendes, PhD

Associate Editor

PLOS Computational Biology

Douglas Lauffenburger

Deputy Editor

PLOS Computational Biology

Reviewer's Responses to Questions

**Comments to the Authors:**

Reviewer #1: The authors have successfully answered all my comments. I think the information provided by this study is relevant and will be very useful for other researchers in the field.

Reviewer #2: The authors did an excellent job addressing both my and the other reviewers concerns. The manuscript is now suitable for publication in PLOS comp. Bio.

**Have the authors made all data and (if applicable) computational code underlying the findings in their manuscript fully available?**

Reviewer #1: Yes

Reviewer #2: Yes

PLOS authors have the option to publish the peer review history of their article (what does this mean?). If published, this will include your full peer review and any attached files.

Reviewer #1: No

Reviewer #2: No

---

## [Editor Report · Acceptance letter]

20 Jun 2022

PCOMPBIOL-D-21-01901R1 

Model-based translation of DNA damage signaling dynamics across cell types

Dear Dr Beltman,

I am pleased to inform you that your manuscript has been formally accepted for publication in PLOS Computational Biology. Your manuscript is now with our production department and you will be notified of the publication date in due course.

With kind regards,

Zsanett Szabo
